# Constructing Grith Eight GC-LDPC Codes Based on the GCD FLRM Matrix with a New Lower Bound

**DOI:** 10.3390/s22197335

**Published:** 2022-09-27

**Authors:** Kun Zhu, Hongwen Yang

**Affiliations:** School of Information and Communication, Beijing University of Posts and Telecommunications, Beijing 100876, China

**Keywords:** full-length row multiplier matrix, greatest common divisor, globally coupled LDPC, large girth

## Abstract

By connecting multiple short, local low-density parity-check (LDPC) codes with a global parity check, the globally coupled (GC) LDPC code can attain high performances with low complexities. The typical design of a local code is a quasi-cyclic (QC) LDPC for which the code length is proportional to the size of circulant permutation matrix (CPM). The greatest common divisor (GCD)-based full-length row multiplier (FLRM) matrix is constrained by a lower bound of CPM size to avoid six length cycles. In this paper, we find a new lower bound for the CPM size and propose an algorithm to determine the minimum CPM size and the corresponding FLRM matrix. Based on the new lower bound, two methods are proposed to construct the GC-QC-LDPC code of grith 8 based on the GCD based FLRM matrix. With the proposed algorithm, the CPM size can be 45% less than that given by sufficient condition of girth 8. Compared with the conventional GC-LDPC construction, the codes constructed with the proposed method have improved performance and are more flexible in code length and code rate design.

## 1. Introduction

Channel coding has always been one of the most important underlying technologies in communication systems. Capacity-achieving codes such as turbo codes, low-density parity-check (LDPC) codes, polar codes, and quantum codes have been proposed together with various near-maximum likelihood(ML)-decoding algorithms such as belief-propagation, list-decoding, guessing random additive noise decoding (GRAND), etc. [1,2,3]. LDPC codes have attracted much attention for its low decoding complexity and excellent performance [4,5,6,7]. The structure of quasi-cyclic (QC) LDPC code is relatively simple, so it is beneficial for hardware implementation. Therefore, QC-LDPC code plays an important role in many communication protocols, including wireless sensor networks [8,9,10]. Since regular LDPC codes have lower error-floor [11] and irregular LDPC codes can be easily obtained by transforming the well-designed regular codes [12,13], the construction of regular full-ank QC-LDPC codes is a very popular topic [14,15,16,17,18].

The full-length row multiplier (FLRM) matrix is a typical regular matrix used in construction of QC-LDPC codes. The FLRM matrix consists of multiple circulant permutation matrices (CPMs) [15,16,17,18] arranged in *J* rows and *L* columns. The exponent matrix of FLRM matrix is derived from the product of row coefficient vector and column indexes. Girth in Tanner graph is one of the most important characters of LDPC codes for it determines the minimum distance of the code. In addition, short cycles will produce trapping sets, stopping sets, and absorption sets, resulting in heavy error-floor and performance degradation [19]. Hence, many scholars devote themselves to improving the girth of the LDPC code [20,21,22]. It has been shown in [17] that the girth of FLRM codes cannot exceed eight, since there exist cycles of length eight regardless of the CPM’s size. In [16], the greatest common divisor (GCD) scheme was proposed to construct the grith 8 FLRM matrix. As the general form of CPM, the affine permutation matrices (APMs) can also adopt the GCD scheme to construct LDPC code of girth 8 [23]. Recently, the authors of [18] optimized the row coefficient’s vector to find the lower bound of CPM size for each pair of (J,L) and further confirmed the validity of the GCD construction method for the FLRM matrix with girth 8.

A globally coupled (GC) LDPC code is a new type of coupled LDPC code proposed by Juane Li [24] where multiple short local LDPC codes are connected with a set of global check nodes. As a result, GC-LDPC can effectively realize a long code with multiple short component codes, avoiding the construction of a completely new longer LDPC code [25]. The GC-LDPC code is adept in correcting erasures clustered in bursts and performs greatly under additive white Gaussian noise (AWGN) channels and binary erasure channels (BECs). With local/global two-phase decoding, the decoders of the component LDPC codes are reusable [26]. In [27], the Reed–Solomon code-based construction of GC-LDPC code was proposed. In [26], the array dispersion was applied to scale the GC-LDPC code, which makes the design of the GC-LDPC code more flexible in code-length selection. With the Reed–Solomon-Like construction [28], local codes and global coupling part of GC-LDPC code can be constructed separately. In addition, the protograph-based GC-LDPC code [29] and tail-biting GC-LDPC code [30] perform well. In [26,31], GC-LDPC codes were used for NAND Flash, and the relative independent structure and local/global two-phase decoding can reduce the critical path and decoding latency. Recently, ref. [25] proposed the free-ride coding to realize an implicit GC-LDPC code, and parallel encoding and efficient decoding algorithms were proposed based on the unique structure of the GC-LDPC code.

Since the existing GC-LPDC code cannot achieve girth 8, in this paper, we consider the construction of a GC-LDPC code based on the FLRM matrix. The main contributions of this paper are as follows:1.We find a new lower bound of CPM size *P* to achieve girth 8 for the GCD-based FLRM matrix and propose an algorithm that can output the minimum *P* and the corresponding FLRM matrix for the given J,L.2.The two new methods for constructing the GC-LDPC code is proposed based on the FLRM matrix with a new lower bound.3.We find that the performance of GC-LPDC is more sensitive to the number of six length cycles than the girth.

The finding of this study is particularly meaningful for the code designer for they will have more freedom in choosing *P*, code length, or code rate. In addition, the simulation results show that the code constructed with proposed method has improved performances than the code constructed with existing methods.

The sections of this paper are organized as follows. In Section 2, we discuss cycles in the FLRM matrix and the new lower bound of CPM’s size *P*. An algorithm is proposed to find the smallest *P* and the corresponding FLRM matrix for a given (J,L). In Section 3, two code construction methods are proposed for the GC-LDPC code based on the GCD-based FLRM matrix. In Section 4, we show the simulation results, and we conclude our paper in Section 5.

## 2. New Lower Bound of CPM Size

The parity check matrix of the QC-LPDC is generated through a CPM of size *P* and an exponent matrix of size J×L. The elements of the exponent matrix E is the product of the row and column coefficients:(1)$$E=\left[\begin{array}{cccc} a_{0}b_{0} & a_{0}b_{1} & \cdots & a_{0}b_{L-1}\\ a_{1}b_{0} & a_{1}b_{1} & \cdots & a_{1}b_{L-1}\\ \vdots & \vdots & \ddots & \vdots \\ a_{J-1}b_{0} & a_{J-1}b_{1} & \cdots & a_{J-1}b_{L-1} \end{array}\right],$$
where $a_{i}$, and $b_{i}$ are integers with $0\leq a_{0}<a_{1}<\cdots <a_{J-1}$ and $0\leq b_{0}<b_{1}<\cdots <b_{L-1}$. The row coefficient vector is $a=(a_{0},a_{1},\cdots ,a_{J-1})$, and the column coefficients vector is $b=(b_{0},b_{1},\cdots ,b_{L-1})$. The column coefficients vector of the FLMR matrix is b=(0,1,2,⋯,L−1). The maximum girth of FLRM codes is eight [17]. The GCD method [16] is an efficient framework for constructing FLRM codes, attaining girth eight.

The girth of GC-LDPC code is essentially the length of the shortest cycle in E. A cycle $W=\{W_{0},W_{1},...,W_{l-1}\}$ in matrix E is a sequence of the elements of E such that [32]
(2)$$\sum_{i=0}^{l/2-1}\left(W_{2i}-W_{2i+1}\right)=0\left(\mathrm{mod}\;P\right),$$
where l∈{4,6,8,⋯} is the length of cycle.

A length of four cycles in E forms a 2×2 sub-matrix of E. Let $E_{4}^{{}^{\prime }}$ be a 2×2 sub-matrix of E, which consists of two distinct rows and two distinct columns of E:(3)$$E_{4}^{{}^{\prime }}=\left[\begin{array}{cc} a_{i}b_{x} & a_{i}b_{y}\\ a_{j}b_{x} & a_{j}b_{y} \end{array}\right],$$
where 0≤i<j<J and 0≤x<y<L. $E_{4}^{\prime }$ can form a length of four cycles if its elements satisfy (Equation 2). This is equivalent to $\det\left(E_{4}^{\prime }\right)=0\left(\mathrm{mod}\;P\right)$. Hence, the condition for E with no four-length cycle is that all 2×2 submatrices of E are non-singular. This condition can be equivalently expressed as follows:(4)$$\left(a_{j}-a_{i}\right)\left(b_{y}-b_{x}\right)\neq 0\left(\mathrm{mod}\;P\right),$$
for all 0≤i<j<J, and 0≤x<y<L.

A length of six cycles lies in a 3×3 sub-matrix of E. Consider the following:(5)$$E_{6}^{{}^{\prime }}=\left[\begin{array}{ccc} a_{i}b_{x} & a_{i}b_{y} & a_{i}b_{z}\\ a_{j}b_{x} & a_{j}b_{y} & a_{j}b_{z}\\ a_{k}b_{x} & a_{k}b_{y} & a_{k}b_{z} \end{array}\right],$$
where 0≤i<j<k<J,0≤x<y<z<L. A path of length 6 in $E_{6}^{\prime }$ satisfying (Equation 2) will define a cycle of length 6. Similarly to (Equation 4), if we define the following:(6)$$\;\begin{array}{c} S_{1}=\left(a_{k}-a_{i}\right)\left(b_{y}-b_{x}\right)+\left(a_{k}-a_{j}\right)\left(b_{z}-b_{y}\right)\\ S_{2}=\left(a_{k}-a_{i}\right)\left(b_{y}-b_{x}\right)+\left(a_{j}-a_{i}\right)\left(b_{z}-b_{y}\right)\\ S_{3}=\left(a_{k}-a_{i}\right)\left(b_{z}-b_{y}\right)+\left(a_{k}-a_{j}\right)\left(b_{y}-b_{x}\right)\\ S_{4}=\left(a_{k}-a_{i}\right)\left(b_{z}-b_{y}\right)+\left(a_{j}-a_{i}\right)\left(b_{y}-b_{x}\right)\\ S_{5}=\left(a_{k}-a_{j}\right)\left(b_{y}-b_{z}\right)+\left(a_{j}-a_{i}\right)\left(b_{y}-b_{x}\right)\\ S_{6}=\left(a_{k}-a_{j}\right)\left(b_{y}-b_{x}\right)+\left(a_{j}-a_{i}\right)\left(b_{y}-b_{z}\right), \end{array}$$
then any of $S_{1},S_{2},\cdots ,S_{6}$ being zero (module *P*) indicates the existence of *P* cycles of length 6. The possible paths of these cycles are illustrated in Figure 1. These cycles can be classified into two types: the ‘L’ type paths correspond to $S_{1},S_{2},S_{3},S_{4}$, and the ‘X’ type paths correspond to $S_{5},S_{6}$.

Since $a_{i}<a_{j}<a_{k}$ and $b_{x}<b_{y}<b_{z}$, all $S_{1},S_{2},S_{3},S_{4}$ are positive and are upper bounded. A sufficiently large *P* can guarantee $S_{i}\left(\mathrm{mod}P\right)=S_{i}$ or $S_{i}\neq 0\left(\mathrm{mod}P\right)$ and, thus, avoid the type ’L’ length 6 cycle.

Even with infinite *P*, ’X’ type cycles may exist for $S_{5}$, or $S_{6}$ may equal to zero. Let $n_{a}=a_{j}-a_{i}$, $m_{a}=a_{k}-a_{j}$, $n_{b}=b_{y}-b_{x}$, and $m_{b}=b_{z}-b_{y}$. $S_{5}$ and $S_{6}$ can be rewritten as follows.
(7a)$$\begin{array}{cc} S_{5} & =-m_{a}\cdot m_{b}+n_{a}\cdot n_{b}, \end{array}$$
(7b)$$\begin{array}{cc} S_{6} & =m_{a}\cdot n_{b}-n_{a}\cdot m_{b}. \end{array}$$

For the ’X’ type length 6 cycles, the condition $S_{5}=0$ or $S_{6}=0$ can be equivalently expressed as the difference ratio as follows.
(8a)$$\begin{array}{cc} \frac{n_{a}}{m_{a}} & =\frac{m_{b}}{n_{b}},\;\mathrm{if}\;S_{5}=0, \end{array}$$
(8b)$$\begin{array}{cc} \frac{n_{a}}{m_{a}} & =\frac{n_{b}}{m_{b}},\;\mathrm{if}\;S_{6}=0. \end{array}$$

The condition for the girth 8 FLRM matrix can be derived from the analysis above, and the conclusion is summarized as (Lemmas 1 and 2 of [16])
(9a)$$\begin{array}{cc} \; & \left(n_{a}+m_{a}\right)/\gcd\left(n_{a},m_{a}\right)\geq L, \end{array}$$
(9b)$$\begin{array}{cc} \; & P\geq \left(a_{J-1}-a_{0}\right)\left(L-1\right)+1. \end{array}$$

Equation (9a) is the necessary and sufficient condition for avoiding the type ’X’ cycle of length 6. However, (9b) is only a sufficient condition to avoid type ’L’ cycle of length 6.

The type ’L’ cycle of length 6 exists if and only if any of $\left\{S_{1},S_{2},S_{3},S_{4}\right\}$ equals 0(modP). For the sake of simplicity, we use $P_{\min}^{*}$ to denote the lower bound in (9b).
(10)$$P_{\min}^{*}=\left(a_{J-1}-a_{0}\right)\left(L-1\right)+1.$$

However, (9b) is only a sufficient condition for avoiding the length 6 cycle. It is possible that, for some $P<P_{\min}^{*}$, all $P<P_{\min}^{*}$ and all $S_{i}\neq 0\left(\mathrm{mod}P\right),\;i\in \left\{1,2,\cdots ,6\right\}$ for all 3×3 submatrices. Thus, the real lower bound is given by the following.
(11)$$P_{\min}^{†}=\underset{P}{\mathrm{argmin}}\left\{S_{i}=0\left(\mathrm{mod}P\right),\;\forall i,\forall E_{6}^{{}^{\prime }}\right\}.$$

With the new lower bound $P_{\min}^{†}$, we can extend condition (9b) to the sufficient and necessary condition, as stated in the following theorem.

**Theorem** **1.**
*The Tanner graph corresponds to the FLRM matrix E and has no ’L’ type cycle of length 6 if and only if the following is the case:*

(12a)
$$\begin{array}{cc} \; & P\geq P_{\min}^{*}=\left(a_{J-1}-a_{0}\right)\left(L-1\right)+1,\\ \; & \mathrm{or} \end{array}$$


(12b)
$$\begin{array}{cc} \; & P_{\min}^{†}\leq P<P_{\min}^{*}\;\mathrm{and}\;S_{i}\left(\mathrm{mod}\right)P\neq 0,\forall i,E_{6}^{{}^{\prime }} \end{array}$$

*for all triples $(a_{i},a_{j},a_{k}),\;0\leq i<j<k<J$.*


We propose the Algorithm 1 shown in the next page to find the minimum. The input of Algorithm 1 is the the size of FLRM matrix. The algorithm starts from constructing an FLRM matrix E using the method stated in [16] and E satisfies (9a). This matrix has no length 6 cycle with $S_{5}=0$ or $S_{6}=0$. Then, we calculate path metrics defined in (Equation 6) for all 3×3 submatrices and record the results in a accumulated vector **Sum** where the $e^{th}$ element, Sum(e), is the number of length 6 paths for which $S_{i}=e$ for some i∈{1,2,⋯,6}. An example for Sum is illustrated in Figure 2 with J=5,L=20.
**Algorithm 1:** Construct E with the minimum $P=P_{\min}^{†}$
**Require:** J,L**Ensure:** E and $P_{\min}^{†}$1:construct $E^{{}^{\prime }}$ satisfing (9a) as [18];2:Initialize $\mathrm{Sum}=0;P^{\mathrm{init}}=a_{J-1}\times \left(L-1\right)$3:**for** (∀i,j,k,x,y,z;0≤i<j<k<J,0≤x<y<z<L) **do**4:   For each 3×3 submatrix, calculate path metric $S_{i},i=1,2,\cdots ,6$ with (Equation 6);5:   **for** (i=0:6) **do**6:     $\mathrm{Sum}(S_{i})++;$7:   **end for**8:**end for**9:**for** ($e=\max\left\{a_{J-1},L-1\right\}:P^{\mathrm{init}}$) **do**10:   Candidate = True11:   **for** ($r=1:P^{\mathrm{init}}/e$) **do**12:     **if** Sum(e×r)≠0 **then**13:        Candidate = False; break;14:     **end if**15:   **end for**16:   **if** Candidate = True **then**17:     $E_{\mathrm{temp}}=E^{{}^{\prime }}$(mod)e18:     **for** (∀i,j,x,y,0≤i<j<J,0≤x<y<L) **do**19:        For each 2×2 submatrix, calculate determinant $D=\det\left(E_{\mathrm{temp},4}^{\left(i,j,x,y\right)}\right)$20:        **if** D=0(mod)e **then**21:          Candidate = False; break;22:        **end if**23:     **end for**24:   **end if**25:   **if** Candidate = True **then**26:     $P_{\min}^{†}=e$; break;27:   **end if**28:**end for**29:$E=E_{\mathrm{temp}}$

If the Tanner graph has no length 6 cycle under CPM size *P*, then there will be no path metric such that $S_{i}=0\left(\mathrm{mod}P\right)$, or $S_{i}=rP$, for i∈{1,2,⋯,6} and r∈{1,2,⋯}. This is reflected in Sum as Sum(P)=Sum(2P)=Sum(3P)=⋯=0. In Algorithm 1, the lines 11∼15 check this condition. Note that if the FLRM matrix contains no length 6 cycle, it does not mean that it contains no length 4 cycle. Thus, in lines 17∼23, we check (Equation 4) for all 2×2 submatrices. The algorithm searches the indices of Sum in range $\max\left\{a_{J-1},L-1\right\}\leq e<P_{\min}^{*}$. The new lower bound $P_{\min}^{†}$ is the minimum index of the zero elements of **Sum**.
(13)$$P_{\min}^{†}=\underset{\begin{array}{c} \max\left\{a_{J-1},L-1\right\}\leq e<P_{\min}^{*},\\ 1\leq r<P_{\min}^{*}/e \end{array}}{\mathrm{argmin}}\left\{\mathrm{Sum}(e\times r)=0\right\}.$$

It is possible that the Tanner graph corresponding to the FLRM matrix has only ’X’ type cycles or only ’L’ type cycles. Combining Theorems 1 and 2, the properties concerning these cases are summarized as the following theorem:

**Theorem** **2.**
*The cycles in the Tanner graph corresponding to FLRM matrix E have the following properties:*
*1*.
*If $P\geq P_{\min}^{*}$ and there exists triple $(a_{i},a_{j},a_{k}),\;0\leq i<j<k<J$ such that*
*$\left(n_{a}+m_{a}\right)/\gcd\left(n_{a},m_{a}\right)<L$, then all cycles of length 6 are type ’X’ cycle.*
*2*.
*If $P<P_{\min}^{*}$ and $\left(n_{a}+m_{a}\right)/\gcd(\left(n_{a},m_{a}\right)\geq L$ for all triples $\left(a_{i},a_{j},a_{k}\right)$, 0≤i<j<k<J, then the number of 6 length cycles equals to Sum(P)×P.*



For index *e*, Sum(e) is the number of length 6 cycles under CPM size P=e. It can be seen from Figure 2 that, as the CPM size increase, the number of length 6 cycles decrease rapidly. Latter in Section IV, we can see that number of length 6 cycles has critical influences on the performance.

Some results on the new lower bound $P_{\min}^{†}$ are shown in Table 1 and Table 2, where $P_{N}$ denotes the number of valid *P* below the original lower bound $P_{\min}^{*}$ in [16]. From these Tables, it can be seen that the new lower bound is significantly smaller than the original one. In cases J=5 and L=13, the new lower bound $P_{\min}^{†}$ is about 40% lower than $P_{\min}^{*}$. In the case of J=6, the reduction is about 45% for most values of *L*. The reduction ratio becomes even larger when *J* and *L* increase.

High-rate LDPC codes require sufficiently large *L*. The lower bound $P_{\min}^{*}$ in [16,18] is roughly in line with $L^{2}\times \left\lfloor (J-1)/2\right\rfloor$ since $a_{2i}>L\times i$ and $a_{J-1}$ are very close to $a_{J-2}$ for even *J*. The code length $N=P\times L\approx L^{3}\times \left\lfloor (J-1)/2\right\rfloor$ grows with $L^{3}$. With the new lower bound $P_{\min}^{†}$, the limitation on *N* can be greatly reduced, which is significant for the construction of high-rate girth-8 QC-LDPC with small FLRM matrix sizes.

Note that when J≤4, we have $P_{\min}^{*}=P_{\min}^{†}$ for all *L*. This is because $a_{J-1}-a_{0}$ is small; hence, the summations of the paths are enough to iterate over all available values under $\left(a_{J-1}-a_{0}\right)\left(L-1\right)+1$.

## 3. Construct Girth-8 GC-LDPC with New Lower Bound

In general, the parity check matrix of the GC-LPDC has the following structure:(14)$$H_{\mathrm{gc}}=\left[\begin{array}{c} \underline{\begin{array}{cccc} H_{L}^{0} & & & \\ & H_{L}^{1} & & \\ & & \ddots & \\ & & & H_{L}^{t-1} \end{array}}\\ \begin{array}{ccc} & & \begin{array}{ccc} & H_{G} & \end{array} \end{array} \end{array}\right],$$
where $H_{L}^{i},i=0,1,\cdots ,t-1$ is the local parity check matrix of *i*th local codeword, and $H_{G}$ is the global parity check that connects all local codes together. The GC-LDPC codeword can be decoded globally with $H_{\mathrm{gc}}$, or it firstly decodes each local codeword with $H_{L}^{i}$ and then decodes the entire codeword with $H_{G}$. Figure 3 illustrates the Tanner graph of GC-LDPC code with a general structure. The circle/square symbol indicates the variable/check nodes, respectively. From Figure 3, it can be seen more intuitively that local codes update and exchange information through the global check node.

### 3.1. Review of the Construction of GC-LDPC Code

The GC-LDPC code is generally derived from the well-designed J×L regular matrix with girth six. The designed matrix is then deformed into the GC-LDPC code using displacement, masking, and dispersion.

In [24], the GC-LDPC code is constructed on GF(*q*), and the matrices are provided by the following: (15)$$\begin{array}{c} B_{1}=\left[\begin{array}{ccccc} \alpha^{0}-1 & \alpha^{}-1 & \cdots & \alpha^{q-3}-1 & \alpha^{q-2}-1\\ \alpha^{q-2}-1 & \alpha^{0}-1 & \cdots & \alpha^{q-4}-1 & \alpha^{q-3}-1\\ \vdots & \vdots & \ddots & \vdots \\ \alpha^{}-1 & \alpha^{2}-1 & \cdots & \alpha^{q-2}-1 & \alpha^{0}-1 \end{array}\right],\\ B_{2}=\left[\begin{array}{cccc} 1 & 1 & \cdots & 1\\ 1 & \beta & \cdots & \beta^{p-1}\\ \vdots & \vdots & \ddots & \vdots \\ 1 & \beta^{p-1} & \cdots & {\left(\beta^{p-1}\right)}^{p-1} \end{array}\right], \end{array}$$
where α is a primitive element over GF(q), $\beta =\alpha^{e}$, and q−1=pe. Any 2×2 submatrix of $B_{1}$ and $B_{2}$ is non-singular, which guarantees that $B_{1}$ and $B_{2}$ are free of length 4 cycles.

A Reed–Solomon code has been used in [27] to construct the GC-LDPC code on GF($2^{s}$). Let α be the primitive element of GF($2^{s}$), $2^{s}-1=c\times n$, *c* is the primitive factor of $2^{s}-1$, and $\gamma =\alpha^{c}$. The elements of vector $s=(1,\gamma ,\gamma^{2},...,\gamma^{n-1})$ are the cyclic elements over GF$\left(2^{s}\right)$ with order *n*. Since the *n*’s smallest prime factor is $p_{s}>d$, $B_{RS}\left(d,n\right)$ given by the following:(16)$$B_{RS}\left(d,n\right)=\left[\begin{array}{cccc} 1 & \gamma & \cdots & \gamma^{n-1}\\ 1 & \gamma^{2} & \cdots & {\left(\gamma^{2}\right)}^{n-1}\\ \vdots & \vdots & \ddots & \vdots \\ 1 & \gamma^{d} & \cdots & {\left(\gamma^{d}\right)}^{n-1} \end{array}\right]$$
and it satisfies the 2×2 constraint (Equation 4). The construction methods above are based on a matrix satisfying the 2×2 constraint, and then we obtain all local codes and global parts using segmentation, masking, and re-organization.

The Reed-Solomon-Like construction is proposed in [28] which makes the design of GC-LDPC code more flexible. The local and global codes can be individually designed as follows: (17)$$\mathrm{RS}\left(a,b,d\right)=\left[\begin{array}{ccccc} \gamma^{0} & \gamma^{a} & \gamma^{2a} & \cdots & \gamma^{a(b-1)}\\ \gamma^{0} & \gamma^{2a} & \gamma^{4a} & \cdots & \gamma^{2a(b-1)}\\ \vdots & \vdots & \vdots & \ddots & \vdots \\ \gamma^{0} & \gamma^{\mathrm{ad}} & \gamma^{2\mathrm{ad}} & \cdots & \gamma^{\mathrm{da}(b-1)} \end{array}\right],$$
where *a* denotes the scaling factor, and *b* and *d* are the numbers of columns and rows of RS(a,b,d).

To avoid the length 4 cycles between global and local parts, the number of rows of local codes and global check should be constrained to $d_{L}<\left\lfloor \frac{b_{L}}{a_{L}}\right\rfloor$ and $d_{G}<\min\left\{a_{L}\right\}$, respectively, where the subscript L and G indicate that the parameter belongs to local codes or global part. With the same set of parameters, the codes derived from above methods have a similar performance. For this reason, only the Reed–Solomon-Like method is used as the comparison scheme since this construction method is more flexible.

### 3.2. GCD-Based GC-LDPC Code

The GC-LDPC can be obtained through the matrices designed above with some operations without changing the property of the cycle. In the following, we will apply the GCD method to construct girth eight GC-LDPC codes.

**Construction** **1.***We generate the girth 8 FLRM matrix through Algorithm 1 for the code with column weight L. The number of rows, $d_{L}^{i},i\in \left[0,t\right)$, and $d_{G}$ is set in accordance to the code rate’s requirement. The first $d_{\mathrm{LMax}}$ rows of the matrix are split into $E_{L}^{i}$, where $d_{\mathrm{LMax}}=\max\left(d_{L}^{i}\right)\leq L-d_{G}$. Finally, $E_{L}^{i}$ is rearranged into diagonal forms, as shown in Figure 4*.

With Construction 1, the code rate is given by $r_{1}≃1-\sum \left(d_{L}^{i}\right)/J$. Increasing $r_{1}$ requires an increase in *J*. Since the lower bound of *P* increases rapidly with the increase in *J*, the code length $N=P_{\min}\times J$ will soon become unacceptable. To address this problem, we provide Construction 2 as follows.

**Construction** **2.***In Construction 2, the FLRM matrix generated via Algorithm 1 is split into two parts, namely*$E_{G}$*and*$E_{L}^{\mathrm{Max}}$*, with size the as*1×J*and*(L−1)×J*, respectively. Then, the two sub-matrices are copied t times and interleaved along columns, as shown in Figure 5II. According to the requirements of each sub-matrix, each interleaved copy is cut, and then global and local sub-matrices are obtained as illustrated in Figure 5II. At last, these matrices are rearranged into the form as illustrated in Figure 5III*.

The rate of this code is $r_{2}=\sum \left(d_{L}^{i}\right)/\left(t\times J\right)$. In case $d_{L}^{i}=L-1$, the code rate is $r_{2}=\left[t\left(L-1\right)+1\right]/\left(t\times J\right)$. The global part is limited to having a $d_{G}=1$ row to ensure that no new cycles occur in the global part.

If necessary, the column cyclic mask can be used in the construction to reduce the column’s weight. Let $m={\left(m_{0},...,m_{J-1}\right)}^{T}$ be the binary cyclic mask vector. The elements of cyclic mask matrix M are given by $M_{i,j}=m_{\left(i+j+c)(\mathrm{mod}\;J\right)},i\in \left[0,J\right),j\in \left[0,L\right)$ where *c* is the randomly selected initial offset. For example, if all local codes have the same size of 5×8, the weight of m is $M_{n}=1$; then, with the randomly generated $m={\left(0,0,1,0,0\right)}^{T}$ and c=0, the mask matrix is constructed as follows:(18)$$M=\left[\begin{array}{cccccccc} 0 & 0 & 0 & 1 & 0 & 0 & 0 & 0\\ 0 & 0 & 0 & 0 & 1 & 0 & 0 & 0\\ 1 & 0 & 0 & 0 & 0 & 1 & 0 & 0\\ 0 & 1 & 0 & 0 & 0 & 0 & 1 & 0\\ 0 & 0 & 1 & 0 & 0 & 0 & 0 & 1 \end{array}\right],$$
where the entries with value ’1’ indicate the positions to be masked in local codes $H_{L}^{i}$ of the same positions.

The focus of the code’s construction is to achieve girth 8. For the decoding of LDPC codes, the short cycles will lead to inappropriate messages passing among nodes or error propagation. Moreover, these short cycles can form the local structure of trapping set and stopping set. All these issues will cause performance degradation. Therefore, the LDPC code should be carefully designed to avoid short cycles.

## 4. Simulation Results

In this section, we use simulations to verify the proposed GC-LDPC code. All codes in this section were simulated in the AWGN channel with BPSK modulations. The min-sum algorithm with a scaling factor 0.55 is used in decoding, and the maximum iteration number of local/global decoding is 50.

We also use the extrinsic information transfer (EXIT) chart to compare the conventional GC-LDPC code and the proposed codes. The mutual information is given by the following:(19)$$I=1-\int_{-\infty }^{\infty }\frac{1}{\sqrt{2\pi \sigma^{2}}}e^{-\frac{{\left(u-\sigma^{2}/2\right)}^{2}}{2\sigma^{2}}}\log\left(1+e^{-u}\right)du,$$
where $\sigma^{2}$ is the variance of the LLR value.

### 4.1. Comparison of Grith 8 and Grith 6 GC-LDPC Codes

In the following, the proposed code is referred to as code1. We use notation code2, code3, and code4 to denote the baseline codes used for the comparison. The details of these codes are as follows.

code1: The construction of code1 begins with Algorithm 1, which outputs the matrix E of size J×L=4×20 and the minimum CPM size of $P=P_{\min}=400$. Then, we perform Construction 2 with the number of local matrices as t=2 and the size of local matrices as 3×20. The girth of code1 is 8, the size of the exponential matrix is 7×40, the code length is $N_{1}=$ 16,000, and the code rate is $r_{1}=0.829$. The column weight of local matrices is $w_{L}=3$, and the column weight of the entire matrix is $w_{w}=4$.

code2: This code is constructed with a Reed–Solomon-Like scheme [28]. **E** comprises three RS(3,131,3) local matrices and one RS(1,393,1) global matrix. Then, the code was slightly changed to match the parameters with code1. Finally, the size of exponential matrix is 7×40, the CPM size as 393, the code length is $N_{2}=$ 15,720, and the code rate is $r_{2}=0.834$.

code3: This code is constructed to observe the relationship between the performance of GC-LDPC code and the number of length 6 cycles. We construct the grith 6 FLRM matrix B with the vector of row indices a=(0,1,2,3) and the vector of column indices b=(0,1,⋯,19). The CPM size is P=400, the size of exponential matrix is 7×40, the code rate is $r_{3}=0.825$, and the code length is $N_{3}=$ 16,000. According to Theorem 2, this code has no length 4 cycle and no ’L’ type length 6 cycle. All 3×3 sub-matrices involve a large number of length 6 cycles of type ’X’. According to the statistics data obtained in simulation, there is a total of $1.08\times 10^{7}$ length 6 cycles in code3.

code4: The matrix E of code4 is constructed with the GCD method as in code1. The CPM size is P=393, the code rate is $r_{4}=0.825$, and the code length is $N_{4}=15720$. This code only has type ’L’ length 6 cycles. It was observed from the results of Sum in Algorithm 1 that the code has $3.2\times 10^{3}$ length 6 cycles.

Figure 6a compares simulated bit error rate (BER) and frame error rate (FER) performance of code1∼code4. From this figure, we can see that the proposed code1 has the best performance, code4 is in the second position, and code3 is the worst. Note that code1, code3, and code4 are all constructed with the GCD method. The difference mainly lies in the number of length 6 cycles, which is 0, $1.08\times 10^{7}$, and $3.2\times 10^{3}$ for code1, code3, and code4, respectively. The comparison in Figure 6a indicates that the performance of the GC-LDPC code is seriously affected by the number of length 6 cycles. The reason that code3 has the worst performance is that this code contains more length 6 cycles than other codes. It is worth noting that although the girth of code1 is 8, while the girth of code2 and code3 is 6, the performance of code1 is only slightly improved compared to the other two. The EXIT charts of codes 1, 2, 4 also indicate that the GCD-based GC-LDPC and Reed–Solomn-Like GC-LDPC have nearly the same performance. An interest observation from Figure 6 is that the dominant factor affecting the error rate’s performance is the number of length 6 cycles rather than the girth. Therefore, the designer does not have to guarantee girth 8. The more important concern should be the number of length 6 cycles. In other words, it may not be necessary to select a *P* corresponds to the zero element of **Sum** in Algorithm 1. A *p* value for which $\sum_{r}\left(\mathrm{Sum}(P\times r)\right)$ is relatively small may be sufficient.

### 4.2. GC-LDPC Code with Minimum Size CPM

In this section, we consider some GC-LDPC codes for which GCD-based FLRM matrices are generated with *L* and a selected from Table 1 and Table 2. The code is constructed with Construction 2 mentioned in the last section, and the column cyclic mask is used to mitigate the column’s weight.

case 1 

J=5



In this case, we consider two codes, code5 and code6, with parameters listed in Table 3.

In the AWGN channel, the best range of column weight is $w_{c}=\left[3,4\right]$. We use the column cyclic mask of $M_{n}=1$ to reduce $w_{c}$ to 3. The masked codes are denoted as code5 mask1 and code6 mask1. The code rate of code5 mask1 is r=0.69, and it is r=0.729 for code6 mask1. Figure 7 indicates that both the original codes and the masked codes have shown good BER/FER performances without an error floor. Note that code 5 has short lengths but improved performances than code 6. This is because code 5 has a smaller code rate. With a code length as large as code 5, the performance is dominated by the code rate.

case 2 

J=6



In this case, the 6×13 FLRM matrix is used to construct code7 with Construction 2. The CPM size is P=199, the number of the local codes is t=3, the code length is N=7761, and the code rate is r=0.591. The column cyclic mask of weight $M_{n}=1$ or 2 is applied, and the masked codes are denoted as code7 mask1 and code7 mask2. The code rate of masked codes are both r=0.59. The simulated error rate performance is shown in Figure 8. This results indicates that the GC-LPDC codes with different column weight requirements can be obtained using the method of GCD-based FLRM matrices and column cyclic masks, and these codes have good performances without ab error floor.

## 5. Conclusions

In this paper, we show that the lower bound of CPM’s size can be even lower than what we known from the previous literature. An algorithm is proposed to find the minimum size of CPM for the GCD-based FLRM matrix. Based on this algorithm, two construction methods were proposed to construct girth 8 GC LPDC codes. In addition, we find that the dominant factor that affects the performance is the number of length 6 cycles rather than the girth. With the proposed algorithm, the CPM size can be 45% less than that given by a sufficient condition of girth 8. Compared with the conventional GC-LDPC construction, the codes constructed with the proposed method have better performances and are more flexible in code length and code rate design.

## Figures and Tables

**Figure 1 sensors-22-07335-f001:**
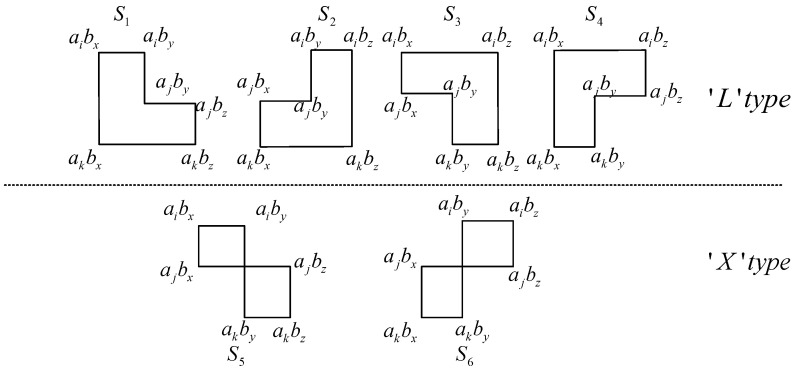
Possible paths of length 6 cycles corresponding to (Equation 6).

**Figure 2 sensors-22-07335-f002:**
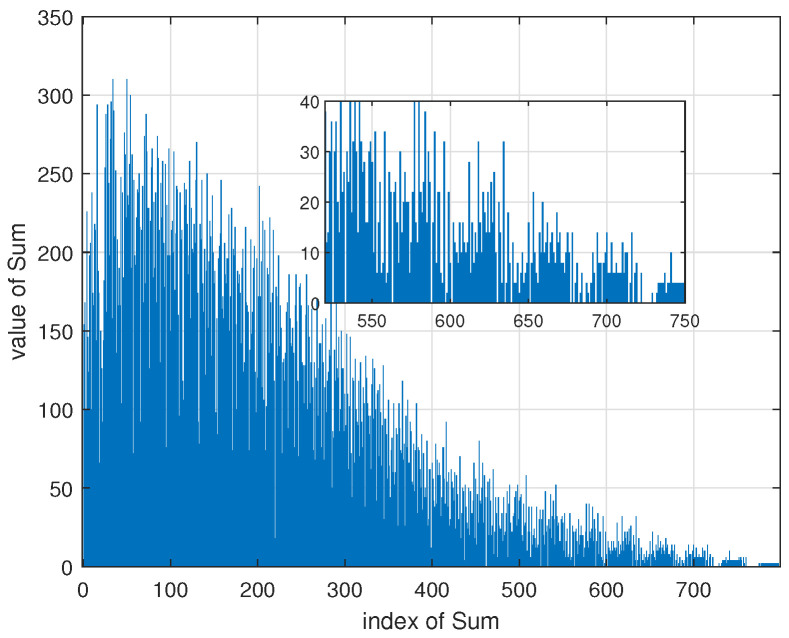
An example for the value of **Sum** for J=5,L=20.

**Figure 3 sensors-22-07335-f003:**
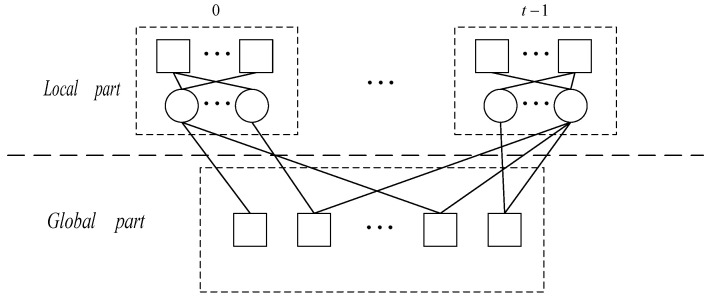
GC-LDPC’s Tanner graph.

**Figure 4 sensors-22-07335-f004:**
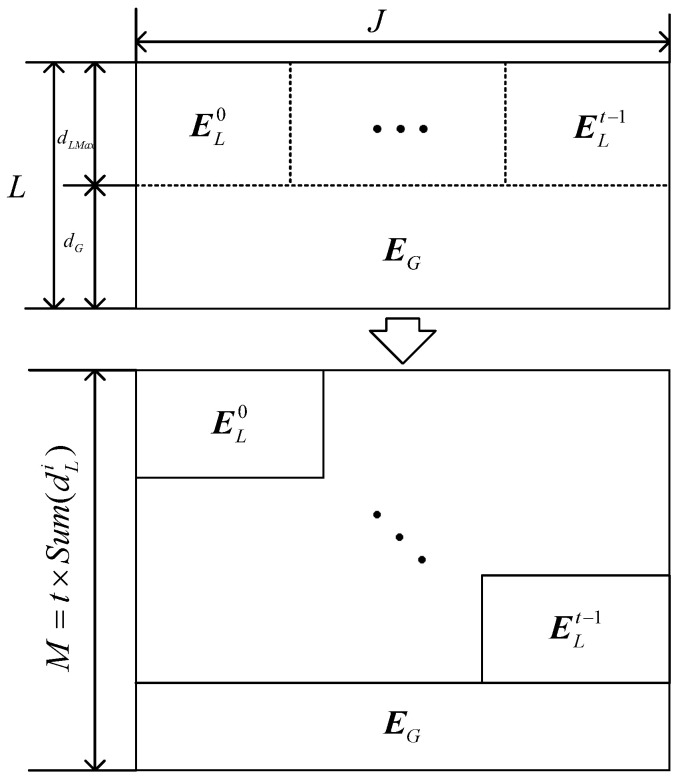
Illustration of construction 1.

**Figure 5 sensors-22-07335-f005:**
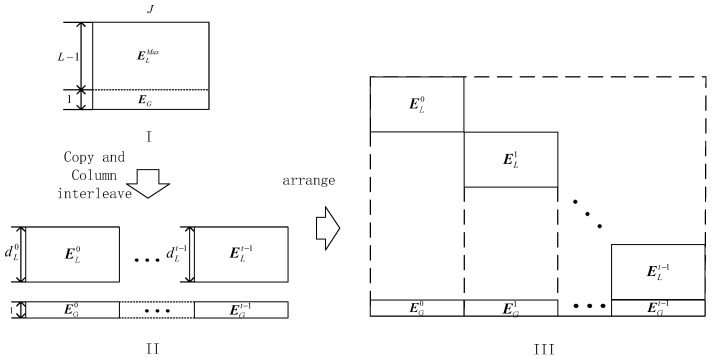
Illustration of Construction 2.

**Figure 6 sensors-22-07335-f006:**
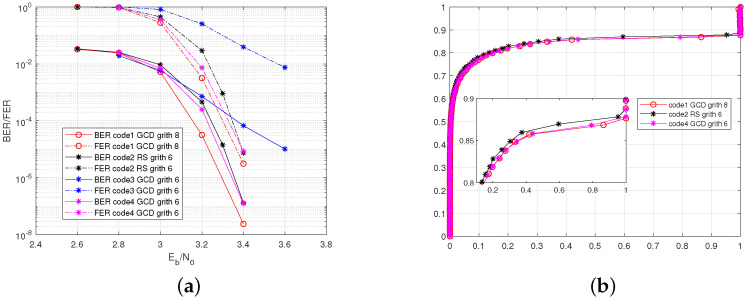
Comparison of the GC-LDPC codes with different numbers of 6-cycles. (**a**) BER/FER performances. (**b**) EXIT chart.

**Figure 7 sensors-22-07335-f007:**
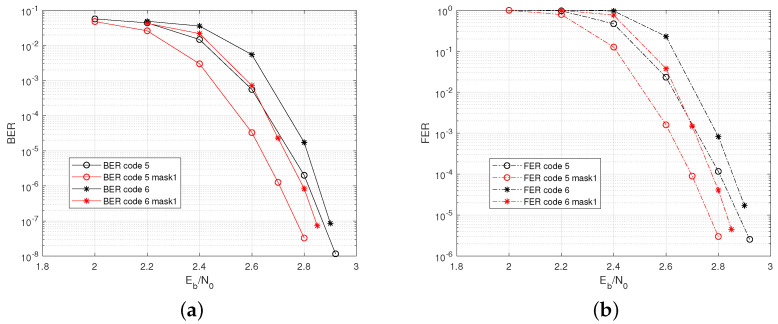
Performance of code5, code6, code5 mask1, and code6 mask1 for J=5. (**a**) BER. (**b**) FER.

**Figure 8 sensors-22-07335-f008:**
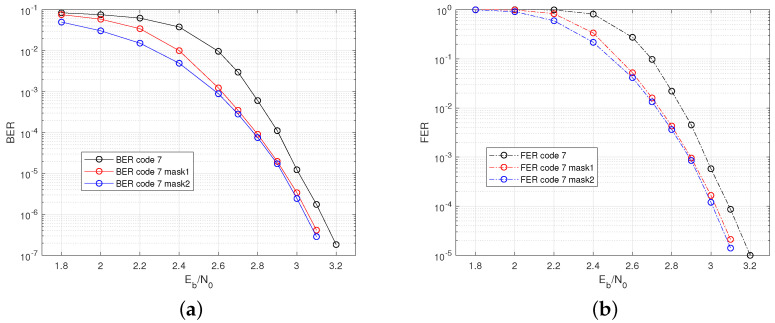
Performance of code7, code7 mask1, and code7-mask2 for J=6. (**a**) BER. (**b**) FER.

**Table 1 sensors-22-07335-t001:** (5, *L*) girth-eight FLRM code with minimum CPM size.

*L*	$a_{0}a_{1}a_{2}a_{3}a_{4}$	$P_{\min}^{*}$ Lower Bound in [18]	$P_{\min}^{†}$ New Lower Bound	$P_{N}$ Number of *P* Below $P_{\min}^{*}$
5	0,1,5,11,12	49	49	0
6	0,1,8,9,14	71	63	1
7	0,2,7,11,16	97	67	5
8	0,1,8,11,18	127	111	4
9	0,2,9,13,20	161	103	9
10	0,1,10,11,23	208	143	8
11	0,1,11,18,23	231	165	8
12	0,1,14,15,26	287	221	10
13	0,2,13,17,28	337	199	21
14	0,1,14,17,30	391	285	19
15	0,1,17,18,32	449	368	16
16	0,1,16,23,33	496	407	15
17	0,1,17,22,35	561	357	21
18	0,1,18,23,37	630	529	21
19	0,1,19,32,39	703	595	21
20	0,1,20,23,42	799	525	41

**Table 2 sensors-22-07335-t002:** (6, *L*) girth-eight FLRM code with minimum CPM size.

*L*	$a_{0}a_{1}a_{2}a_{3}a_{4}a_{5}$	$P_{\min}^{*}$ Lower Bound in [18]	$P_{\min}^{†}$ New Lower Bound	$P_{N}$ Number of *P* Below $P_{\min}^{*}$
6	0,2,7,11,16,18	91	63	7
7	0,2,7,11,16,18	108	67	8
8	0,1,8,11,18,19	134	134	0
9	0,2,9,13,20,22	176	103	13
10	0,1,10,11,23,24	217	217	0
11	0,1,11,14,24,25	251	251	0
12	0,2,13,17,28,30	331	199	24
13	0,2,13,17,28,30	361	199	26
14	0,1,14,17,30,31	404	315	1
15	0,2,15,19,32,34	477	259	34
16	0,5,16,23,34,39	586	357	49
17	0,1,17,22,38,39	625	399	4
18	0,1,18,23,40,41	698	501	6
19	0,2,19,23,40,42	757	403	53
20	0,1,20,23,42,43	818	693	1

**Table 3 sensors-22-07335-t003:** The parameters of code 5 and 6.

	J×L	*P*	*N*	*r*	*t*
code5	5×14	285	11,970	0.692	3
code6	5×16	407	19,536	0.731	3

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
