# Peer review of "Constructing Grith Eight GC-LDPC Codes Based on the GCD FLRM Matrix with a New Lower Bound"

_sensors, 2022, doi:10.3390/s22197335_

Round 1
Reviewer 1 Report
The paper proposes the GC-LDPC with 8 girth. The author should be clearer regarding the position of novelty or the paper between the state of the art. Below are my comments
1) The abstract is lack of the brief results sentences
2) Write again the full form of LDPC in the introduction when it appears for the first time in line 14
3) Clearly define the math symbol of J and L in line 22. It is not clear and not to directly related between the symbols and definitions in Line 23
4) Clearly state in line 53 that there is no one has proposed the construction of GC-LDPC with FLRM matrix
5) Punctuate (7a) and (7b)
6) Make cd in (9a) as subscript
7) Fill the blank space in page 4
8) Put { } for the set of number, for example i = {1, 2, …, 6}
9) = Sum(3P)· · · = 0 -> = Sum(3P) = · · · = 0
10) line 11~15 -> lines 11~15
11) Could you provide the factor graph illustration of GC-LDPC?
12) Typo in Theorem 2 point 1: ((na, ma)
13) Shift line 134 to left
14) B_1 in (15) seems strange. It is shown as 4 rows. Check the second column of B_1. If yes, then the vertical dots are not needed
15) 2 lines below line 132: what it cn? Is it c_n or c x n? define n directly in the same sentence
16) Do not punctuate the sentence before (16), instead, punctuate (16)
17) Minimum 3 sentences to form one paragraph. Revise line 136 – 138
18) Shift to the left line 139
19) Add more discussion the superiority of girth 8 and 6 and why in the simulation result
20) Table 1 and Table 2 -> Tables 1 and 2
21) Analyze code 5 with rate lower than code 6 has better performance in Figure 6
22) More finding claim should be mentioned in the conclusion
23) Add more explanation of the equation in deriving the EXIT Chart
24) Why the comparison of other technique is only with reed-solomon? Is there no other technique? Explain in the paper
25) Is there any complexity analysis?
26) Provide more application example in the real world for example, will this paper contribute to future technology such as 6G? Discuss the position other codes such as Polar Code, Quantum Code, GRAND decoding, etc.
Author Response
Thank you for your valuable amendments. These revisions have indeed effectively improved the quality of the article. There were some writing lapses and verbal inappropriacies in the previous editions. Next show the changes in the order you suggest.

Reviewer 2 Report
Manuscript is technically sound. However, a lot of grammatical corrections are required. For eg.
Abstract: To achieve girth 8, the size of CPM should not less 4 than the lower bound. This sentence requires correction.
Introduction: The quasi-cyclic (QC) LDPC code, which is hardware friendliness,....This sentence requires correction.
"Base on this new lower bound, two methods....". This sentence requires correction.
Sec. 3.2 Construction2 should be written as Construction 2. Same correction in sec. 4.1 and 4.2
Abstract needs to be explained in a better way. It is difficult to understand what exactly the author wants to communicate.
Authors have presented a novel contribution but they have not take good care of the presentation and grammatical importance. Entire article must be revised in terms of presentability, uniformity in the notations and grammatical correctness.
Author Response

(The authors gave the same response as above.)

Round 2
Reviewer 1 Report
The authors have addressed all my comments except the position of the code for 5G and beyond such as 6G. As we know LDPC dan Polar Code are used in 5G. Also other happening code such as quantum ldpc code, and grand universal code. I think this information can enhance the introduction part for enrich the knowledges.
